# Alcohol Consumption and Risk of Rheumatoid Arthritis among Chinese Adults: A Prospective Study

**DOI:** 10.3390/nu13072231

**Published:** 2021-06-29

**Authors:** Hannah VanEvery, Wenhao Yang, Nancy Olsen, Le Bao, Bing Lu, Shouling Wu, Liufu Cui, Xiang Gao

**Affiliations:** 1Department of Nutritional Sciences, The Pennsylvania State University, State College, PA 16802, USA; hqv5028@psu.edu; 2Department of Rheumatology and Immunology, Kailuan General Hospital, Tangshan 063000, China; ywhywhywh123@163.com (W.Y.); cuiliufu@hotmail.com (L.C.); 3Division of Rheumatology, Department of Medicine, Penn State Milton S. Hershey Medical Center, Hershey, PA 17033, USA; nolsen@pennstatehealth.psu.edu; 4Department of Statistics, The Pennsylvania State University, State College, PA 16802, USA; lebao@psu.edu; 5Department of Medicine, Harvard Medical School, Boston, MA 02115, USA; blu1@bwh.harvard.edu; 6Department of Cardiology, Kailuan General Hospital, Tangshan 063000, China; drwusl@163.com

**Keywords:** rheumatoid arthritis, prospective, cohort, alcohol, epidemiology

## Abstract

Alcohol consumption may be associated with the risk of rheumatoid arthritis (RA), but potential sex-related differences in this association have not been explored. Thus, we utilized 87,118 participants in the Kailuan Study, a prospective cohort initiated in 2006 to study the risk factors of cardiovascular disease in a Chinese population. We included those that did not have RA at baseline (2006), and performed cox proportional hazard modeling to calculate the hazard ratio (HR) and 95% confidence interval (95% CI) of RA according to the levels of alcohol consumption (never or past, light or moderate (<1 serving/day for women, <2 servings/day for men), and heavy (>1 serving/day for women, >2 servings/day for men), adjusting for age, sex, body mass index, and smoking. Diagnoses of RA were confirmed via medical record review by rheumatologists. From 2006 to 2018, we identified 87 incident RA cases. After adjusting for potential confounders, the HR of RA was 1.26 (95% CI: 0.62, 2.56) for participants with light or moderate alcohol consumption and 1.98 (95% CI: 0.93, 4.22) for participants with heavy alcohol consumption) versus non-drinkers. The HR of each 10 g increase in alcohol consumption was 1.11 (95% CI: 0.98, 1.26) (*p*-trend = 0.09). A significant association between alcohol consumption and RA risk was observed in women, but not in men (*p* for interaction = 0.06). Among women, each 10 g increase in alcohol consumption was significantly associated with a high risk of RA (HR: 1.56; 95% CI: 1.06, 2.29). In contrast, each 10 g increase in alcohol consumption was not significantly associated with the risk of RA in men (HR: 1.10; 95% CI: 0.97, 1.25). Excluding past drinkers generated similar results. In this prospective Chinese cohort, increasing alcohol consumption was associated with an elevated risk of RA among women, but not in men. These findings highlight the importance of incorporating analysis of sex differences into future studies of alcohol consumption and RA risk.

## 1. Introduction

Approximately 60% of the risk of rheumatoid arthritis (RA) can be attributed to genetic factors [1,2], but environmental and lifestyle risk factors for RA have also been identified, such as smoking and silica dust exposure [3,4]. The impact of alcohol consumption on RA risk remains unclear. A previous meta-analysis based on eight prospective studies suggested that the impact of alcohol consumption on RA risk might follow a J-shaped curve, and people with low-to-moderate alcohol consumption may have a lower RA risk, relative to non- and heavy-drinkers [5]. However, this analysis did not separate past drinkers from never drinkers, which may have impacted the results. Individuals may stop drinking due to the presence of certain underlying conditions, related to RA. Furthermore, there may be an interaction that has not been fully explored, such as between alcohol consumption and other factors (e.g., sex), that may affect the association between alcohol consumption and RA risk [6,7,8].

Female sex is a well-established risk factor for RA [9]. The association between alcohol consumption and RA risk may be different in men and women because the effect of alcohol on estrogen signaling [10] and the hypothalamic-pituitary-adrenal (HPA) axis [11,12], which may be associated with RA risk [13,14]. To date, only one case-control study reported associations between alcohol consumption and RA risk stratified by sex [15]. Importantly, alcohol consumption by sex is likely to vary across cultures, and the sex-specific consumption pattern of Asian participants is different from the European or North American consumption patterns [16]. However previous investigation of the association between alcohol consumption and RA risk has been primarily conducted utilizing European-descendent populations. 

Thus, we conducted this study using a large Chinese prospective cohort to test whether alcohol consumption is associated with altered risk of developing RA in men and women.

## 2. Materials and Methods

### 2.1. Study Population

This study included participants in an ongoing prospective cohort, the Kailuan Study, which began recruiting in 2006 and is being conducted in Tangshan City, China. In total, 101,510 participants (between 18 and 98 years old; 81,110 men and 20,400 women) were enrolled in the study at baseline in 2006. Between June 2006 and October 2007, each participant completed a baseline survey, referred to here as the “2006 survey”, that collected information on demographics, lifestyle factors, medical conditions, and medication usage [17,18,19]. After the baseline visit, the survey was re-administered every two years. Biennially and at baseline, laboratory tests and physical examinations were carried out on the participants. New health events were documented on an annual basis. The current analysis includes 87,118 participants (19,100 women and 68,018 men; mean age = 51.8 years) that did not have an RA diagnosis at baseline and have complete data on alcohol consumption. The participant flow chart can be found in Appendix A.

### 2.2. Assessment of Alcohol Consumption

Data on alcohol consumption were collected via a questionnaire, as previously described [20]. In this questionnaire, participants were asked whether they consumed alcoholic beverages in the past 12 months and, if so, which type of beverage (beer, wine, and hard liquor). Participants were further asked to specify the amount and frequency of intake. The grams of alcohol consumed per day were calculated by multiplying the average frequency of consumption (times per day) by the typical amount consumed of each type of beverage and its average ethanol content (5.0 g per 100 g beer, 12.0 g per one glass of wine, and 40.0 g per serving of hard liquor). One standard drink was defined as approximately 15 g of ethanol. The validity of the self-reported alcohol consumption data is supported by previous analyses of the Kailuan Study, which demonstrated a dose–response relationship between alcohol consumption and high-density lipoprotein cholesterol (HDL-C) concentrations in the participants [20]. Participants were further classified into groups based on the sex-specific alcohol consumption groups described in the dietary guidelines: never or past consumption of alcohol, light-to-moderate consumption (women: <1 servings/day; men: <2 servings/day), and heavy consumption (women: >1.0 servings/day; men: >2 servings/day) [20,21]. 

### 2.3. Assessment of Rheumatoid Arthritis Diagnoses

A team of rheumatologists reviewed the medical records of possible RA cases to confirm each diagnosis of RA, as detailed elsewhere [19]. In short, potential cases of RA were found via querying the Municipal Social Insurance Institution in China, which included all participants of the Kailuan Study. After a potential case of RA was found among the participants of the Kailuan Study, three rheumatologists reviewed the medical records to confirm the RA diagnosis. This ensured that each confirmed case of RA met the classification criteria set out by the American College of Rheumatology/European League Against Rheumatism [22]. This study exclusively contains rheumatologist-confirmed cases of RA.

### 2.4. Assessment of Covariates

At baseline and biennially, fasting blood draws were performed to quantify total high-sensitivity c-reactive protein (hs-CRP), cholesterol, triglycerides, HDL-C, low-density lipoprotein cholesterol (LDL-C), and glucose. An autoanalyzer (Hitachi 747, Hitachi, Tokyo, Japan) in the central laboratory of Kailuan General Hospital was used for these measurements.

In addition to the blood draws, at baseline and every two years, trained nurses assessed the anthropometric characteristics of each participant (e.g., blood pressure, height, and weight) as previously described [18]. The height and weight measurements were used to calculate body mass index (BMI) by dividing weight in kilograms by height squared in meters. Additionally, during the baseline and biennial visits, participants self-reported data on baseline age, sex, birth place, smoking, and medical history (e.g., cardiovascular diseases, and current medications such as antihypertensive and lipid-lowering agents) via a questionnaire [23].

### 2.5. Statistical Analysis

In this study, a two-sided *p* < 0.05 was considered statistically significant, and the analyses were performed using SAS, version 9.3 (SAS Institute, Inc., Cary, NC, USA).

We used Cox proportional hazard models to investigate the association between baseline alcohol consumption (never or past, light-to-moderate (women: 0–1.0 servings/d; men: 0–2.0 servings/d), and heavy (women: >1.0 serving/d; men: >2 servings/d)) and RA risk, after adjusting for potential confounders, including age, sex, BMI (<23, 23–27.5, and >27.5 kg/m^2^), and smoking (never/past, occasional/daily). In a sensitivity analysis, we further adjusted for hs-CRP (<1, 1–3, and ≥3 mg/L), diabetes (diabetic, pre-diabetic, and non-diabetic), hypertension (hypertension, prehypertension, and no hypertension), HDL-C (quartiles), LDL-C (tertiles), and triglycerides (tertiles), although we were aware that these factors could be intermediate variables in the alcohol-RA pathway. We used the “never or past” alcohol consumption group as the reference group. The person-time of each participant was calculated using the time from the baseline visit to the end of follow-up on 31 December 2016, death, or time of diagnosis of RA, whichever came first. The proportional hazards assumption was satisfied for the Cox models. The significance of the trend was tested by assigning the median value (in grams) of alcohol consumption in each of the alcohol groups. 

To test the effect of liquor, beer, and wine consumption on RA risk, we utilized a Cox model, including the same covariates mentioned previously. Wine and beer were combined due to the low number of participants that reported consuming either.

The effect of biological sex on the association between alcohol consumption and RA risk is not fully understood, and it is possible that the impact of alcohol consumption is additive in relation to the impact of biological sex. Thus, we also tested the additive interaction between alcohol consumption and biological sex on the future risk of RA [24]. 

We also investigated whether BMI (<23, 23–27.5, and >27.5 kg/m^2^), smoking (current, yes vs. no), hs-CRP, LDL-c, HDL-c, triglycerides, or total cholesterol affected the association between alcohol consumption and RA risk using the likelihood ratio test, adjusting for the previously mentioned covariates. We further repeated the main Cox model analyses to investigate the association between baseline alcohol consumption and seropositive RA (including both rheumatoid factor (RF) and anti-citrullinated protein antibody (a-CCP)-positive cases), seronegative RA, RF-positive RA, RF-negative RA, a-CCP-positive RA, and a-CCP-negative RA. 

Past, but not current, alcohol consumption may be associated with changes in future risk in RA [25]. Thus, we conducted a sensitivity analysis by excluding past drinkers, adjusting for age, sex, BMI, and smoking status.

To understand the potential impact of unmeasured confounding on the association between alcohol intake and RA risk, we calculated E-values [26]. E-values measure how strong (in risk ratio scale) an association would have to be between an unmeasured confounder and both the exposure (e.g., alcohol intake) and outcome (e.g., RA risk), to entirely explain the alcohol–RA relation [26].

## 3. Results

During the 10 years of follow-up, 87 incident cases of RA were identified. Of the participants included in this study, 29.8% reported current drinking at baseline (light or moderate consumers: 17.3%; heavy consumers: 12.4%). At baseline, among men, 37.2% of participants reported current alcohol consumption, while among women, 3.27% of participants reported current alcohol consumption. Alcohol consumption was associated with current smoking, presence of hypertension, high concentrations of triglycerides, LDL-cholesterol, and HDL-cholesterol (Table 1).

The multivariate-adjusted HR of RA was 1.26 (95% CI: 0.62, 2.56) for participants with light or moderate alcohol consumption and 1.98 (95% CI: 0.93, 4.22) for participants with heavy alcohol consumption) compared to non-drinkers. The HR of each 10 g increase in alcohol consumption was 1.11 (95% CI: 0.98, 1.26) (*p*-trend = 0.09) (Table 2). Further adjustment for hs-CRP, diabetes, hypertension, LDL-c, HDL-c, and triglycerides produced similar results (Table 2). 

There was a significant association between alcohol consumption and RA risk in women, but not in men (*p* for interaction = 0.06) (Figure 1A). The HR for each 10 g increase in alcohol consumption was 1.56 (95% CI: 1.06, 2.29) in women and 1.10 (95% CI: 0.97, 1.25) in men (Figure 1A). The E-values were 2.49 and 1.43 for women and men, respectively. Excluding past drinkers (Figure 1B) or further adjusted for diabetes, hypertension, hs-CRP, LDL-c, HDL-c, and triglycerides generated similar results (data not shown). 

There was no significant association between liquor, or beer and wine, consumption and RA risk (Appendix A). We did not observe a significant interaction between alcohol consumption and smoking, BMI, hs-CRP, LDL-c, HDL-c, triglycerides, or total cholesterol in relation to RA risk (P_interaction_ > 0.10 for all). There was also no significant association between alcohol consumption and any RA subtypes (Appendix A).

## 4. Discussion

In this large, prospective cohort study, we demonstrated that irrespective of smoking status and other potential risk factors for RA, alcohol consumption was associated with a high risk of RA among women, but not in men. These results remained consistent after restricting the reference group to only those that reported no previous alcohol consumption.

To date, of the 10 prospective studies exploring alcohol consumption and RA risk, the majority (6 of 10) found no significant association between alcohol consumption and RA risk [8,15,27,28,29,30], whereas others found that moderate alcohol consumption was associated with a lower risk of RA [7,25,31,32]. One nested case-control study that matched on sex, reported sex-specific analyses of the association between alcohol consumption and RA risk [15]. This study found no significant association between alcohol consumption and RA risk in their cohort as a whole, or in men or women separately [15]. To the best of our knowledge, ours is the first cohort study to investigate the potential sex differences in alcohol consumption and RA risk. Although the most recent meta-analysis reported sex-specific analyses [5], their results might have been impacted by the cohort effect due to none of the included studies reporting age-specified results. In addition to the prospective cohort and nested case-control studies, one mendelian randomization study was performed [33], which found no association between 24 alcohol-associated single-nucleotide polymorphisms and seropositive RA. These results should be interpreted with caution because this study did not report sex-specific results. The impact of genetic factors on alcohol intake is different between participants of different sexes and cultural backgrounds. 

In addition to prospective studies, cross-sectional research has found that alcohol consumption is inversely associated with both RA risk and disease activity [34,35,36,37,38,39]. Although these studies add valuable insights to the literature, they may have been affected by reverse causality. Evaluating alcohol consumption prior to RA diagnosis, in a prospective cohort, mitigates this risk. However, future studies are warranted to prospectively investigate the potential roles of alcohol on the disease prognosis among RA patients. Some of the heterogeneity in the prospective literature may be explained by the methods used to measure alcohol intake, varying use of confounding variables in the analyses, and the populations utilized in the prospective cohorts. In the current study, as with the majority of previous cohort studies on this topic, participants were asked in a questionnaire whether they consumed alcoholic beverages in the past 12 months and, if so, which type of beverage (beer, wine, and hard liquor), how often it was consumed, and how much was typically consumed [20]. From these questionnaires, alcohol consumed per day or per week was calculated, and then groups were formed using various cut-offs to calculate the risk of RA associated with different alcohol consumption groups. Although this method is useful, it does not allow for differentiating between binge drinking and regular drinking, or daily consumption. The physiological effects of binge drinking are markedly different from moderate daily alcohol consumption [40], and it is possible that the association between alcohol consumption and RA risk differs by drinking behavior (binge versus regular consumption). Furthermore, the relationship between alcohol consumption and RA risk may be affected by a variety of confounding factors. Whereas smoking and certain genetic factors have been identified as factors that affect the relationship between alcohol consumption and RA risk [6,7], it is possible that there are cultural or socioeconomic factors that can also impact the association between alcohol consumption and RA. Accounting for confounding factors has been inconsistent in previous prospective cohort studies. Some studies adjusted for only age [27], only age and smoking [32], or only education level and smoking [15], whereas others adjusted for many anthropometric and lifestyle factors [25,29,30]. 

Our study is the first cohort study, to the best of our knowledge, to investigate the association between alcohol consumption and RA risk in a Chinese population. In contrast, previous studies are mainly based on Western populations. Thus, it is important to consider the genetic differences in alcohol metabolism between European-descendent and Asian populations. In Asian populations, a variation in the allele for aldehyde dehydrogenase 2 causes impaired alcohol clearance, and an accumulation of the alcohol breakdown product acetaldehyde in the body, even after consuming moderate amounts of alcohol [41]. Thus, for the same amount of alcohol consumed, the physiological effects of alcohol may be amplified in Asian populations compared to European-descendant populations. This allele variant is present in as much as 45% of the population (depending on exact location within Asia) [42]. Because we did not have genotype data, it is possible that the presence of this variant may have confounded our results. In this study, we utilized cutoff points for the alcohol consumption categories that are commonly applied to European-descendant populations [21]. Thus, it is possible that if participants had the variant of aldehyde dehydrogenase 2, the effect of alcohol on the risk of RA may be larger than it would be in a European population.

Sex differences in the metabolism and impact of alcohol may underlie the sex-specific association between alcohol consumption in women and risk of RA. Women tend to have higher blood alcohol concentrations than men for the same alcohol consumed, even after accounting for body weight [43]. Additionally, for women, alcohol levels in the blood rise faster and stay elevated longer compared to men [44]. Alcohol consumption is further known to effect estrogen signaling in the body, and just 10–15 g/day (one drink) was associated with a higher risk of breast cancer in women [10]. Although being a female sex is associated with a higher risk of RA, the exact effect of estrogen on RA risk is not clear [13]. Further research is needed to understand the connection between estrogen and RA risk, and potentially the link between alcohol consumption and estrogen. Alcohol also affects the HPA axis [11,12]. Chronic alcohol exposure can cause dysregulation of the HPA axis, and there is evidence that the effect of alcohol on the HPA axis is more pronounced in those of female sex [11]. Previous research has linked polymorphisms in the receptor for glucocorticoids, an important regulator of the HPA axis, with the risk of RA [14], but more work is needed to elucidate the mechanisms that may connect alcohol, the HPA axis, and RA risk. Further, the microbiome may mediate the sex differences in RA risk that may be associated with alcohol consumption. Both acute and chronic alcohol consumption affect the microbiome [45], dysbiosis was noted in patients with [46], and sex differences in the microbial content of the gut were previously identified [47]. More research is needed to understand the role of the microbiome in the pathway potentially connecting alcohol consumption and RA risk.

The strengths of this study include the large number of participants and rheumatologist-confirmed diagnoses of RA. To the best of our knowledge, this is also the first prospective cohort study to investigate the association between alcohol consumption and RA risk in a cohort that is not majority European or European-descendent. 

Our study has several limitations. The small number of incident RA cases limited the power for detecting small-to-moderate effect sizes. However, the prevalence of RA in the studied population is in accordance with current estimates of the prevalence of RA in China, as detailed previously [19]. The generalizability of the findings is limited because the participants in this cohort represent one racial group and are not representative of other races or socioeconomic classes. It is also possible that alcohol consumption was not accurately captured, especially for heavy drinkers, because these data were collected via self-report. Underreporting could lead to an underestimation of the impact of heavy alcohol consumption on RA risk. Although this type of underreporting is possible for every participant, it is more likely in women as social drinking is acceptable for men in Chinese culture, but less acceptable for women [48]. Although we did account for several confounding factors in our analyses, we cannot exclude the possibility that other, unmeasured factors (e.g., drinking behavior and overall diet quality) affected the association between alcohol consumption and RA risk. As this was a secondary analysis of a prospective cohort that was originally designed to investigate cardiovascular disease, some risk factors for RA (e.g., silica dust exposure) were not quantified. We also did not account for estrogen replacement therapy, as this was not included in the questionnaires until 2010, but we do not expect that this impacted our results as the prevalence of estrogen replacement therapy was low in Kailuan community (in 2014, <100 women reported using it) [20]. However, a relatively high E-value suggests that the significant association between alcohol intake and RA risk in women is less likely to be explained by an unmeasured factor—the magnitude of the association of this unmeasured factor with both alcohol and RA would have to be large (adjusted risk ratio > 2.4 after adjustment for covariates in the model). 

## 5. Conclusions

In conclusion, our study demonstrates that higher levels of total alcohol consumption in this Chinese population may be associated with a greater risk of RA among women, but not in men. These findings highlight that potential sex differences may exist in the association of alcohol consumption and RA risk, and that further research on this topic is needed. Future prospective studies, in a variety of races and including both sexes, are necessary to further elucidate the relationship between total alcohol consumption and RA risk. Further, prospective studies should consider including measures of specific alcoholic beverages (e.g., beer and wine), so that the relationship between the consumption of each type of beverage and the risk of RA may be explored.

## Figures and Tables

**Figure 1 nutrients-13-02231-f001:**
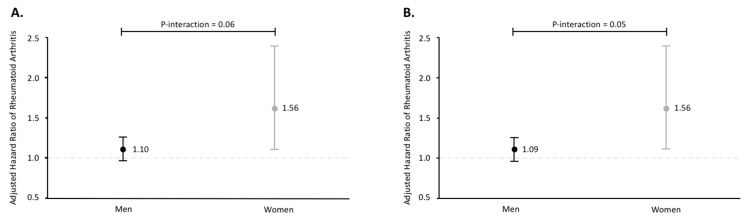
Sex-specific hazard ratio of rheumatoid arthritis for each 10 g increase in alcohol consumption in all participants (**A**) and excluding past drinkers (**B**), adjusting for age, body mass index (<18.5, 18.5–23, 23–27.5, and >27.5 kg/m^2^), and smoking (never, past, and current).

**Table 1 nutrients-13-02231-t001:** Baseline characteristics in 2006 by alcohol consumption group among 87,118 Kailuan participants without rheumatoid arthritis, adjusted for age and sex ^a^.

	Alcohol Consumption Group		
	Never or Past	Light or Moderate	Heavy
N	61,161	15,113	10,844
Women, %	30.2	3.88	0.35
Age, year	51.7 ± 0.05	45.5 ± 0.11	49.7 ± 0.13
Alcohol intake, grams/day	0	4.47 ± 0.09	61.8 ± 0.10
Smoking status, number %			
Never	80.9	26.8	12.9
Past	4.35	9.88	6.03
Current	14.7	63.3	81.1
CRP ^b^, mg/L	2.37 ± 0.03	2.48 ± 0.06	2.15 ± 0.07
BMI ^c^, kg/m^2^	24.9 ± 0.02	25.00 ± 0.03	24.7 ± 0.04
Diabetes status, %			
Normoglycemic	71.2	69.4	64.1
Pre-diabetic	18.9	22.8	27.3
Diabetic	9.91	7.80	8.55
Hypertension status, %			
Normotensive	19.1	25.4	13.5
Pre-hypertensive	48.1	47.8	49.6
Hypertensive	32.8	26.8	36.9
LDL-C, mmol/L ^d^	2.27 ± 0.00	2.44 ± 0.01	2.49 ± 0.01
HDL-C, mmol/L ^e^	1.57 ± 0.00	1.53 ± 0.00	1.64 ± 0.00
Triglycerides, mmol/L	1.60 ± 0.01	1.59 ± 0.01	1.75 ± 0.01

^a^ Continuous variables are presented as the mean ± standard error, adjusted for sex and age. Alcohol consumption is grouped into never or past, light-to-moderate (women: 0–1.0 servings/d; men: 0–2.0 servings/d), and heavy (women: >1.0 serving/d; men: >2 servings/d). All variables are age- and sex-adjusted, except age, which is only adjusted for sex. ^b^ C-reactive protein. ^c^ Body mass index. ^d^ Low-density lipoprotein cholesterol (LDL-C). ^e^ High-density lipoprotein cholesterol (HDL-C).

**Table 2 nutrients-13-02231-t002:** Adjusted hazard ratio and 95% confidence interval for risk of rheumatoid arthritis by baseline alcohol consumption group ^a^.

	Alcohol				
	Never or Past	Light or Moderate	Heavy	For Each 10 g Increase in Consumption	P_trend_
# of case/population	62/61161	12/15113	13/10844		
Incidence rate (/10,000 person-years)	1.27	0.94	1.46		
Crude hazard ratio	1.00 (Ref.)	0.73 (0.39, 1.35)	1.17 (0.65, 2.14)	1.04 (0.93, 1.15)	0.49
Sex- and age-adjusted hazard ratio	1.00 (Ref.)	1.08 (0.56, 2.08)	1.64 (0.87, 3.12)	1.09 (0.98, 1.22)	0.11
Multivariate-adjusted hazard ratio ^b^	1.00 (Ref.)	1.26 (0.62, 2.56)	1.98 (0.93, 4.22)	1.11 (0.98, 1.26)	0.09
Sensitivity analyses					
Further adjusted hazard ratio ^c^	1.00 (Ref.)	1.30 (0.64, 2.65)	2.14 (0.99, 4.62)	1.13 (0.99, 1.28)	0.07
Excluding past drinkers ^d^	1.00 (Ref.)	1.15 (0.56, 2.39)	1.78 (0.82, 3.86)	1.10 (0.97, 1.24)	0.15

^a^ Alcohol consumption is grouped into never or past, light-to-moderate (women: 0–1.0 servings/d; men: 0–2.0 servings/d), and heavy (women: >1.0 serving/d; men: >2 servings/d). ^b^ Adjusted for sex, age, body mass index (<18.5, 18.5–23, 23−27.5, and >27.5 kg/m^2^), and smoking (never, past, and current). ^c^ Further adjusted for hs-CRP (<1, 1−3, and ≥3 mg/L), diabetes (non-diabetic, pre-diabetic, and diabetic), hypertension (no hypertension, pre-hypertension, and hypertension), HDL-C (quartiles), LDL-C (tertiles), and triglycerides (tertiles). ^d^ Past drinkers excluded from the reference group, and model adjusted for sex, age, body mass index (<18.5, 18.5–23, 23−27.5, and >27.5 kg/m^2^), and smoking (never, past, and current).

## Data Availability

Data described in the manuscript, code book, and analytic code will be made available upon request by contacting Drs. Cui and Gao.

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
