# Peer review of "Alcohol Consumption and Risk of Rheumatoid Arthritis among Chinese Adults: A Prospective Study"

_nutrients, 2021, doi:10.3390/nu13072231_

Round 1
Reviewer 1 Report
This is a hot topic, so with relevance for publication. However, rheumatoid arthritis is more prevalent in women and therefore the validity of the research question is doubtful. Moreover, different kinds of alcoholic beverages can have very different effects and it was not considered in the discussion. For example, the effect of phenolic rich beverages on inflammatory and autoimmune diseases is well known but not considered here.
The novelty of this study is that the sample is a Chinese population. However, this population has a polymorphism in ethanol dehydrogenase, with lower enzymatic activity, so this point must be considered in the discussion. This is specially relevant because the authors use the same amount of alcohol consumption as surveys for non-oriental populations. In the Chinese population, alcohol is more toxic and this fact is not considered in the study. The conclusion should be revised because the authors drew conclusions about alcohol consumption, not about alcoholic beverages. The different matrix of alcoholic beverages should be considered. In addition, the influence of alcoholic beverages on gut microbiota is well known, and the relation between gut microbiota dysbiosis and rheumatoid arthritis has also been recently recognized. This should also be included in the discussion particularly given the differences in gut microbiota between men and women.
Author Response
Thank you for your thoughtful comments and suggestions. We have addressed your suggestions as follows:
This is a hot topic, so with relevance for publication. However, rheumatoid arthritis is more prevalent in women and therefore the validity of the research question is doubtful. Moreover, different kinds of alcoholic beverages can have very different effects and it was not considered in the discussion. For example, the effect of phenolic rich beverages on inflammatory and autoimmune diseases is well known but not considered here.
The novelty of this study is that the sample is a Chinese population. However, this population has a polymorphism in ethanol dehydrogenase, with lower enzymatic activity, so this point must be considered in the discussion. This is specially relevant because the authors use the same amount of alcohol consumption as surveys for non-oriental populations. In the Chinese population, alcohol is more toxic and this fact is not considered in the study.
Thank you for bringing up this very good point. We have added the following to the Discussion section, on the top of page 8:
Starting on line 300, page 8
“In Asian populations, a variation in the allele for aldehyde dehydrogenase 2 causes impaired alcohol clearance, and an accumulation of the alcohol breakdown product acetaldehyde in the body, even after consuming moderate amounts of alcohol [35]. Thus, for the same amount of alcohol consumed, the physiological effects of alcohol may be amplified in Asian populations compared to European-descendant populations. This allele variant is present in as much as 45% of the population (depending on exact location within Asia) [42]. Because we did not have genotype data, it is possible that the presence of this variant may have confounded our results. In this study, we utilized cutoff points for the alcohol consumption categories that are commonly applied to European-descendant populations [21]. Thus, it is possible that if participants had the variant of aldehyde dehydrogenase 2, the effect of alcohol on the risk of RA may be larger than it would be in a European population.”
The conclusion should be revised because the authors drew conclusions about alcohol consumption, not about alcoholic beverages.
Thank you for bringing this to our attention. We have added the following to the beginning of the Discussion section:
Starting on line 369, page 9
“In conclusion, our study demonstrates that higher levels of total alcohol consumption in this Chinese population may be associated with a greater risk of RA among women, but not in men.”
Then, at the end of the discussion section we have added this:
Starting on line 375, page 9
”Further, prospective studies should consider including measures of specific alcoholic beverages (e.g. beer, wine), so the relationship between the consumption of each type of beverage and the risk of RA can be explored.”
The different matrix of alcoholic beverages should be considered.
Thank you for calling our attention to this. We did not test individual alcoholic beverages because of the small number of incident RA cases and did not have sufficient statistical power to detect meaningful association between each individual alcoholic beverage and RA risk. Nevertheless, we have added this to the revised manuscript, based on the reviewer’s suggestion:
Starting on line 138, page 4 (Method section)
“To test the effect of liquor, beer, and wine consumption on RA risk, we utilized a Cox model, including the same covariates as mentioned previously. Wine and beer were combined due to the low number of participants that reported consuming either.”
We have then added this to the results section:
Starting on line 185, page 4
“There was no significant association between liquor, or beer and wine, consumption and RA risk (Supplementary Table S1).”
We have also added a Supplementary Table S1 and added the following to the discussion:
Starting on line 375, page 9
”Further, prospective studies should consider including measures of specific alcoholic beverages (e.g. beer, wine), so the relationship between the consumption of each type of beverage and the risk of RA can be explored.”
In addition, the influence of alcoholic beverages on gut microbiota is well known, and the relation between gut microbiota dysbiosis and rheumatoid arthritis has also been recently recognized. This should also be included in the discussion particularly given the differences in gut microbiota between men and women.
Thank you for bringing up this insightful point. We have added the following to the Discussion section:
Starting on line 327, page 8
“Further, the microbiome could mediate the sex-differences in RA risk that may be associated with alcohol consumption. Both acute and chronic alcohol consumption effect the microbiome(39), dysbiosis has been noted in patients with RA(40), and sex-differences in the microbial content of the gut have previously been identified(41). More research is needed to understand the role of the microbiome in the pathway potentially connecting alcohol consumption and RA risk.”
Reviewer 2 Report
Manuscript ID: nutrients-1218207 Manuscript Title: Alcohol consumption and risk of rheumatoid arthritis: a prospective study Overview: This interesting study demonstrates that higher levels of alcohol consumption in this Kailuan Study of the Chinese population may be associated with a greater risk of RA among women. However, this was not the case not in men. These are interesting and important findings, highlighting the potential contribution of race and sex differences in the association of alcohol consumption and RA risk. Most of the studies have been done in Caucasian populations so far this is the first study that has been done in a Chinese population. Specific comments: 1. The title is clear and concise. 2. The abstract is very well written but it would be good to have a very simple sentence at the beginning of the abstract outlining the aims and objectives of the study, even though it was prospective one. 3. The keywords are appropriate. 4. The introduction is well written and provides the necessary background information. 5. The methods are clearly and comprehensively outlined. 6. The results are well presented, including the tables and figures. I think the authors have done the very best that they could in terms of presentation of complex data in the tables. 7. This reviewer is wondering whether the format of the graphs in figure one might be improved. 8. The discussion is very well written and nicely referenced. 9. Conclusion section is concise and this review agrees that future work should focus on other populations and not just Caucasians. 10. Reference list is complete and citations are appropriate. The authors should be congratulated on a very nice piece of work. This paper will only require very minor revisions.Author Response
Thank you for your thoughtful comments and suggestions. We have addressed your suggestions as follows:
Specific comments:
- The title is clear and concise.
- The abstract is very well written but it would be good to have a very simple sentence at the beginning of the abstract outlining the aims and objectives of the study, even though it was a prospective one.
Thank you for bringing our attention to this. We have added the following sentence to the abstract to outline the aims of the original prospective study:
Starting on line 15, page 1
“Thus, we utilized 87,118 participants in the Kailuan Study, a prospective cohort initiated in 2006 to study risk factors of cardiovascular disease in a Chinese population.”
- The keywords are appropriate.
- The introduction is well written and provides the necessary background information.
- The methods are clearly and comprehensively outlined.
- The results are well presented, including the tables and figures. I think the authors have done the very best that they could in terms of presentation of complex data in the tables.
- This reviewer is wondering whether the format of the graphs in figure one might be improved.
Thank you for the advice. We have updated the figure. We have shortened the Y axis to bring the “men” and “women” labels closer to their respective hazard ratios, and we have increased the difference in shade between the men and women markers, to ensure the reader knows they are looking at sex-specific results in Figure 1.
- The discussion is very well written and nicely referenced.
- Conclusion section is concise and this review agrees that future work should focus on other populations and not just Caucasians.
- Reference list is complete and citations are appropriate. The authors should be congratulated on a very nice piece of work. This paper will only require very minor revisions.
Thank you!
Reviewer 3 Report
I reviewed a paper entitled “Alcohol consumption and risk of rheumatoid arthritis: a prospective study” by Hannah VanEvery et al.
The authors concluded:
“our study demonstrates that higher levels of alcohol consumption in
this Chinese population may be associated with a greater risk of RA among women, but
not in men”.
The current analysis includes 87,118 participants (19,100 women and 68,018 men; mean age=51.8 y).
I think that this paper is well idea, well written with interesting but controversy results.
In fact several study demonstred that alcohol consumption its inverse association with the risk of developing RA.
In addition previous meta-analysis investigating the protective effect of alcohol on developing RA showed that alcohol intake was inversely associated with ACPA-positive RA, proposing a protective effect (Maxwell, J. R., Gowers, I. R., Moore, D. J. & Wilson, A. G. Alcohol consumption is inversely associated with risk and severity of rheumatoid arthritis. Rheumatology (Oxford) 49, 2140–2146 (2010).
Källberg, H. et al. Alcohol consumption is associated with decreased risk of rheumatoid arthritis: results from two Scandinavian case-control studies. Ann. Rheum. Dis. 68, 222–227 (2009).
Bergman, S., Symeonidou, S., Andersson, M. L., Söderlin, M. K. & Barfot Study Group. Alcohol consumption is associated with lower self-reported disease activity and better health-related quality of life in female rheumatoid arthritis patients in Sweden: data from BARFOT, a multicenter study on early RA. BMC Musculoskelet. Disord. 14, 218 (2013).
Lu, B. et al. Associations of smoking and alcohol consumption with disease activity and functional status in rheumatoid arthritis. J. Rheumatol. 41, 24–30 (2014).
Di Giuseppe, D., Alfredsson, L., Bottai, M., Askling, J. & Wolk, A. Long term alcohol intake and risk of rheumatoid arthritis in women: a population based cohort study. BMJ 345, e4230 (2012).
Scott, I. C. et al. The protective effect of alcohol on developing rheumatoid arthritis: a systematic review and meta-analysis. Rheumatology (Oxford) 52, 856–867 (2013).
In more recent meta-analysis (Exploring the effect of alcohol on disease activity and outcomes in rheumatoid arthritis through systematic review and meta-analysis. Published on Scientific Reports 18 May 2021) alcohol consumption is associated with lower disease activity and self-reported health assessment in rheumatoid arthritis.
Data of Swedish study showed that there was an association between alcohol consumption and better health related quality of life in women but not in men (Bergman, S.et al Alcohol consumption is associated with lower self-reported disease activity and better health-related quality of life in female rheumatoid arthritis patients in Sweden: data from BARFOT, a multicenter study on early RA. BMC Musculoskelet. Disord. 14, 218 (2013).
I think that the authors have explain only part of the possible causes.
Many risk factors have been implicated in the development of RA and their impact varies according to patients’ RF and antibodies to citrullinated protein antigen (ACPA) status.
The authors reported that "At baseline and biennially, fasting blood draws were performed to quantify total high-sensitivity c-reactive protein (hs-CRP), cholesterol, triglycerides, HDL-C, low-den#sity lipoprotein cholesterol (LDL-C), and glucose.
Only this parameters? Why?
I not saw ACPA, RF, all patients had ACPA or/and RF positive?
explain please.
In addition the title must be modify specifying the Chinese population.
The discussion section must be enrich.
Author Response
Thank you for your thoughtful comments and suggestions. We have addressed your suggestions as follows:
I think that this paper is well idea, well written with interesting but controversy results.
In fact several study demonstred that alcohol consumption its inverse association with the risk of developing RA.
In addition previous meta-analysis investigating the protective effect of alcohol on developing RA showed that alcohol intake was inversely associated with ACPA-positive RA, proposing a protective effect (Maxwell, J. R., Gowers, I. R., Moore, D. J. & Wilson, A. G. Alcohol consumption is inversely associated with risk and severity of rheumatoid arthritis. Rheumatology (Oxford) 49, 2140–2146 (2010).
Källberg, H. et al. Alcohol consumption is associated with decreased risk of rheumatoid arthritis: results from two Scandinavian case-control studies. Ann. Rheum. Dis. 68, 222–227 (2009).
Bergman, S., Symeonidou, S., Andersson, M. L., Söderlin, M. K. & Barfot Study Group. Alcohol consumption is associated with lower self-reported disease activity and better health-related quality of life in female rheumatoid arthritis patients in Sweden: data from BARFOT, a multicenter study on early RA. BMC Musculoskelet. Disord. 14, 218 (2013).
Lu, B. et al. Associations of smoking and alcohol consumption with disease activity and functional status in rheumatoid arthritis. J. Rheumatol. 41, 24–30 (2014).
Di Giuseppe, D., Alfredsson, L., Bottai, M., Askling, J. & Wolk, A. Long term alcohol intake and risk of rheumatoid arthritis in women: a population based cohort study. BMJ 345, e4230 (2012).
Scott, I. C. et al. The protective effect of alcohol on developing rheumatoid arthritis: a systematic review and meta-analysis. Rheumatology (Oxford) 52, 856–867 (2013).
In more recent meta-analysis (Exploring the effect of alcohol on disease activity and outcomes in rheumatoid arthritis through systematic review and meta-analysis. Published on Scientific Reports 18 May 2021) alcohol consumption is associated with lower disease activity and self-reported health assessment in rheumatoid arthritis.
Data of Swedish study showed that there was an association between alcohol consumption and better health related quality of life in women but not in men (Bergman, S.et al Alcohol consumption is associated with lower self-reported disease activity and better health-related quality of life in female rheumatoid arthritis patients in Sweden: data from BARFOT, a multicenter study on early RA. BMC Musculoskelet. Disord. 14, 218 (2013).
Thank you for providing these citations. They have all been included in the discussion, and we have added the following text:
Starting on line 268, page 7
“In addition to prospective studies, cross-sectional research has found that alcohol consumption is inversely associated with both RA risk, and disease activity (34-39). While these studies add valuable insights to the literature, they may be affected by reverse causality. Evaluating alcohol consumption prior to RA diagnosis, in a prospective cohort, mitigates this risk. However, future studies are warrant to prospectively investigate the potential roles of alcohol on the disease prognosis among RA patients”
I think that the authors have explain only part of the possible causes. Many risk factors have been implicated in the development of RA and their impact varies according to patients’ RF and antibodies to citrullinated protein antigen (ACPA) status. The authors reported that "At baseline and biennially, fasting blood draws were performed to quantify total high-sensitivity c-reactive protein (hs-CRP), cholesterol, triglycerides, HDL-C, low-den#sity lipoprotein cholesterol (LDL-C), and glucose.
Only this parameters? Why?
This is a good point, thank you. The Kailuan Study was originally designed to investigate risk factors for cardiovascular disease, not rheumatologic or autoimmune diseases. To keep within the primary research questions, and to keep within budget, only the blood parameters we mentioned were quantified. There was not justification at the beginning of cohort (2006), or budget, to perform microbial or metabolomic-type analyses, or collect dietary data. We have added the following to the discussion section:
Starting at line 350, page 8
“As this was a secondary analysis of a prospective cohort that was originally designed to investigate cardiovascular disease, some risk factors for RA (e.g. silica dust exposure) were not quantified.”
I not saw ACPA, RF, all patients had ACPA or/and RF positive? explain please.
Thank you for bringing this up. We have added the analyses to the revised manuscript. However, the results should be interpreted with caution given the small number to incident RA cases.
The methods section:
Starting on line 148, page 4
“We further repeated the main Cox model analyses to investigate the association between baseline alcohol consumption and seropositive RA (including both rheumatoid factor (RF) and anti-citrullinated protein antibody (a-CCP) -positive cases), seronegative RA, RF-positive RA, and a-CCP-positive RA.”
Then, we added the following to the results section:
Starting on line 188, page 5
“There was also no significant association between alcohol consumption and any RA subtypes (Supplementary Table S2).”
We have also added a Supplementary Table S2.
In addition the title must be modify specifying the Chinese population.
Thank you for the guidance. The title has been updated to read:
“Alcohol consumption and risk of rheumatoid arthritis among Chinese adults: a prospective study”
The discussion section must be enrich.
Thank you for this feedback. We have added the following to the discussion:
Starting line 307, page 8
“In this study, we utilized cutoff points for the alcohol consumption categories that are commonly applied to European-descendant populations(21). So, it is possible that if participants had the variant of aldehyde dehydrogenase 2, the effect of alcohol on the risk of RA may be larger than it would be in a European population.”
Starting line 327, page 8
“Further, the microbiome could mediate the sex-differences in RA risk that may be associated with alcohol consumption. Both acute and chronic alcohol consumption effect the microbiome(45), dysbiosis has been noted in patients with (46), and sex-differences in the microbial content of the gut have previously been identified(47). More research is needed to understand the role of the microbiome in the pathway potentially connecting alcohol consumption and RA risk.”
We added the following to the Conclusion:
Starting line 375, page 9
“Further, prospective studies should consider including measures of specific alcoholic beverages (e.g. beer, wine), so the relationship between the consumption of each type of beverage and the risk of RA can be explored.”
Round 2
Reviewer 3 Report
the authors have response on all my comment improving their manuscript.